# Identification of Key Metabolic Pathways and Biomarkers Underlying Flowering Time of Guar (*Cyamopsis tetragonoloba* (L.) Taub.) via Integrated Transcriptome-Metabolome Analysis

**DOI:** 10.3390/genes12070952

**Published:** 2021-06-22

**Authors:** Elizaveta Grigoreva, Alexander Tkachenko, Serafima Arkhimandritova, Aleksandar Beatovic, Pavel Ulianich, Vladimir Volkov, Dmitry Karzhaev, Cécile Ben, Laurent Gentzbittel, Elena Potokina

**Affiliations:** 1Information Technologies and Programming Faculty, ITMO University, 197101 St. Petersburg, Russia; L.Grigoreva@gmail.com (E.G.); abeatovic@itmo.ru (A.B.); 2Institute of Forest and Natural Resources Management, Saint Petersburg State Forest Technical University, 194021 St. Petersburg, Russia; vol-j@mail.ru (V.V.); e.potokina@yahoo.com (E.P.); 3Sirius University of Science and Technology, 354340 Sochi, Russia; karzhaevd@gmail.com; 4Vavilov All-Russian Institute of Plant Genetic Resources, 190000 St. Petersburg, Russia; serafima.teplyakova@mail.ru; 5All-Russian Research Institute of Agricultural Microbiology, 196608 St. Petersburg, Russia; p.ulianich@gmail.com; 6Skolkovo Institute of Science and Technology, 121205 Moscow, Russia; C.Ben@skoltech.ru (C.B.); L.Gentzbittel@skoltech.ru (L.G.)

**Keywords:** transcriptome-metabolome integration, differentially expressed genes, gene network analysis, systems biology

## Abstract

Guar (*Cyamopsis tetragonoloba* (L.) Taub.) is an annual legume crop native to India and Pakistan. Seeds of the plant serve as a source of galactomannan polysaccharide (guar gum) used in the food industry as a stabilizer (E412) and as a gelling agent in oil and gas fracturing fluids. There were several attempts to introduce this crop to countries of more northern latitudes. However, guar is a plant of a short photoperiod, therefore, its introduction, for example, to Russia is complicated by a long day length during the growing season. Breeding of new guar varieties insensitive to photoperiod slowed down due to the lack of information on functional molecular markers, which, in turn, requires information on guar genome. Modern breeding strategies, e.g., genomic predictions, benefit from integration of multi-omics approaches such as transcriptome, proteome and metabolome assays. Here we present an attempt to use transcriptome-metabolome integration to understand the genetic determination of flowering time variation among guar plants that differ in their photoperiod sensitivity. This study was performed on nine early- and six delayed-flowering guar varieties with the goal to find a connection between 63 metabolites and 1,067 differentially expressed transcripts using Shiny GAM approach. For the key biomarker of flowering in guar myo-inositol we also evaluated the KEGG biochemical pathway maps available for *Arabidopsis thaliana*. We found that the phosphatidylinositol signaling pathway is initiated in guar plants that are ready for flowering through the activation of the phospholipase C (*PLC*) gene, resulting in an exponential increase in the amount of myo-inositol in its free form observed on GC-MS chromatograms. The signaling pathway is performed by suppression of myo-inositol phosphate kinases (phosphorylation) and alternative overexpression of phosphatases (dephosphorylation). Our study suggests that metabolome and transcriptome information taken together, provide valuable information about biomarkers that can be used as a tool for marker-assisted breeding, metabolomics and functional genomics of this important legume crop.

## 1. Introduction

Guar (*Cyamopsis tetragonoloba* (L.) Taub.) seeds serve as a rich source of a galactomannan polysaccharide (guar gum) that is commonly used in food, cosmetic and oil industry worldwide. Guar is a traditional legume crop of India and Pakistan, used as cattle feed and green manure and that can be eaten as a green bean. Due to the high commercial value of guar gum, many attempts have been made in the last few decades to introduce this short-day legume crop to the countries in more northern latitudes. Attempting this, researchers repeatedly reported a common issue – short day plants had delayed flowering under conditions of long photoperiod which led to late seed maturation and, finally, to a significant loss of yield (e.g., [1]).

Dissection of genes involved in flowering time of legume species is a complex task due to a large number of genetic factors required for the initiation of the generative phase. At least ten genes/QTLs are reported controlling the transition to flowering and maturation in soybeans [2,3,4,5] and still no obvious progress in the cloning of these genes has been made. Previously, we reported that the floral bud formation in guar may be triggered by one certain length of daylight, but flowering *per se* (bud opening) – by another one. Moreover, the onset of flowering time also depends on genetic factors related to the speed of seed germination and the formation of the first true leaf [6]. Thus, focusing not just on the flowering phenotype *per se,* but rather on the biochemical or metabolic landscape that accompanies transition to flowering in guar can be useful in searching for genes responsible for flowering time in this legume species.

Recently, we assessed the delayed flowering of guar under long daylight duration conditions in terms of metabolome profiling [7]. A population of 96 guar genotypes segregating for date of flowering due to different photoperiod sensitivity, was grown under long daylight duration conditions. Sixty-five metabolites showed a significantly higher abundance in early flowering genotypes in comparison to the late flowering genotypes. Among them, seven molecules were suggested as possible biomarkers of early flowering: their ~10-fold increased concentrations in leaf tissues of early flowering guar plants consistently indicated upcoming flowering. Among those key metabolites were liquiritigenin, tetronic and cinnamic acid molecules, and also two inositol isomers (chiro-inositol, myo-inositol) known to play a crucial role in early stages of plant embryogenesis [8].

In the present study we aimed to develop a systems biology approach in combining metabolome profiling results with transcriptome information. First, we carried out a supplementary metabolome profiling experiment and compared the obtained results. Next, we focused on 63 identified metabolites profiles for an integrative study. We reveal genes whose differential expression in early and late flowering genotypes is related to the dramatic difference in concentrations of key metabolites associated with the onset of flowering of guar under long day conditions. The metabolic pathway related to myo-inositol appears to be a key component of the determination of delayed flowering under long-days in this originally short-day species. This may provide targets for genome-assisted breeding to broaden the cultivation area of guar to higher latitudes.

## 2. Materials and Methods

### 2.1. Study Design and Sample Collection

Five guar lines, including three early flowering and two delayed flowering lines on the long day conditions, were selected for the study (Table 1). Seeds of each line were collected each from a single plant during a field experiment in the Krasnodar region of Russia. The plants were derived from five different accessions of the Vavilov Institute of Plant Genetic Resources (VIR) collection. In 2018, an experiment was carried out to grow seeds of each line under conditions of a long photoperiod in a greenhouse (St. Petersburg region, 59°53′39″N) [6]. In 2019, the experiment was replicated with the aim of performing integrative profiling of the metabolome and transcriptome (this study). 

For all of the plants, the date of appearance of seedlings, first true leaf, first floral bud and the date of the first bud opening were recorded. The genotype was recorded as “early flowering” if it turned to the first floral bud formation within 42 ± 7.5 days from the first true leaf appearance. Correspondingly, a genotype was assigned to the “delayed flowering” group if it switched to flowering after 90 ± 0.1 days [7]. 

For further analysis of the metabolome and transcriptome, tissues of the third leaves (the vegetative development phase that precedes flowering in guar) were collected from three different plants per line listed in Table 1. For each plant, the terminal leaflet of the third leaf was laterally separated into two halves – one for the metabolite extraction and another for RNA isolation.

### 2.2. Metabolites Extraction, Derivatization, Identification and Statistical Analysis of Differentially Expressed Metabolites between Groups of Lines with Contrasted Flowering Time under Long Day Conditions

Metabolites were extracted from the leaf tissue using as previously described for guar [9]. Leaflets’ halves were weighed and frozen in liquid nitrogen immediately after harvesting in the greenhouse. Storage of samples was carried out at the temperature of −80 °C. Metabolites of guar leaves were extracted in cold methanol in 1.5 mL Eppendorf type microtubes (SSI, Lodi, CA, USA) for 1 h at +4 °C. Extract solution was transferred to clear Eppendorf microtubes and evaporated using a vacuum concentrator (Labconco, Kansas City, MI, USA).

Metabolome profiling of guar genotypes was performed as described earlier [7]. Derivatization was carried out by sialylation. For this purpose, dry metabolites were dissolved in 20 µl internal standard tricosane (nC23, Sigma, St. Louis, MI, USA) in pyridine solution (1 µg/µl). Silylation was performed using 50 µl N,O-Bis (trimethylsilyl) trifluoroacetamide (BSTFA, Sigma). GC-MS analysis of the samples was performed with a gas chromatograph system (Agilent 6850, Santa Clara, CA, USA) coupled with a mass-spectrometer (Agilent 5975B). The system used a DB-5HT capillary column coated with 5% cross-linked diphenyl (30 m × 250 µm inner diameter, 0.25 µm film thickness; Agilent J&W). 0.8 µm aliquot of the sample was added in spitless mode. Helium was used as a carrier gas. The flow of the front inlet purge was 1 mL/min. Original temperature was set to 70 °C. The temperature increased from 70 °C to 340 °C at the speed of 4 °C/min. Temperature of 250 °C was used for injection. Full-scan mode of the mass spectrometry data was 50 *m/z* – 800 *m/z* at a rate of 2 spectra scans per second. The chromatogram recording was performed on the signal of the total ion current by Agilent ChemStation software. For GC-MS-analysis, three technical replicates of each genotype were used. Calculation of concentration value for each detected metabolite was performed by averaging values from all available replicates, taking into account a value of relative standard deviation (RSD) [10,11].

Peak detection and measurement of integrated areas of peaks were carried out with UniChrome 5.0.19.1162 (www.unichrom.com Accessed on 15 June 2021). Calculation of relative concentration of metabolites normalized by weight of the sample and concentration of the internal tricosane standard (1 µg/µl) was performed by methods of semi-quantitative analysis. Identification of metabolites was performed with Automated Mass Spectral Deconvolution and Identification System AMDIS 32 (http://www.amdis.net/ Accessed on 15 June 2021) using the NIST/EPA/NIH 08 Mass Spectral Library (http://www.nist.gov/srd Accessed on 15 June 2021) and a database of mass spectrometric information, created at the Komarov Botanical Institute. Then, results (10 largest peaks and Retention Index (RI)) were verified by comparison with Golm Metabolome Database (http://gmd.mpimp-golm.mpg.de/analysisinput.aspx Accessed on 15 June 2021). A metabolite was considered identified if Match factor values exceeded a threshold of 700.

Multivariate statistical processing of metabolomic data was carried out using the online analysis platform MetaboAnalyst 4.0 (http://www.metaboanalyst.ca Accessed on 15 June 2021) [12]. The data has been subjected to log transformation (generalized logarithm transformation or glog) and pareto scaling (mean-centered and divided by square root of standard deviation of each variable). Preprocessing of data for multivariate analysis included also imputation of missing values. Missing values were replaced by half of the minimum positive value in the original data. Data filtering was not performed.

### 2.3. RNA Extraction, Library Construction, Sequencing

RNA extraction was performed using the RNeasy Plant Mini kit (Qiagen, Hilden, German). For library preparation RNA integrity was checked with Qubit and Bioanalyzer 2000 (reagents RNA Pico 6000, Agilent, Santa Clara, CA, USA). RNA samples with RIN ≥ 7 were selected for library preparation. Subsequent procedures of library preparation were carried out based on PolyA selection with the reagent kit NEBNext Poly(A) mRNA Magnetic Isolation Module (New England Biolabs, Ipswich, Massachusetts, USA) and NEBNext Ultra II RNA Directional for PCR enrichment. High-throughput sequencing was performed on an Illumina (San Diego, CA, USA) NovaSeq6000 machine with an SP cell for single-end reads of 100 bp length.

### 2.4. Reads Quality Control

MultiQC V. 1.8. software was used for raw sequencing data quality control [13]. Trimmomatic V. 0.39 [14] software with following parameters: Phred 33, leading:30, trailing:30, sliding window:6:30, minlen 40 was applied to remove low quality reads and adapters from raw reads. Illumina random sequencing error correction was performed with Rcorrector V 1.0.3. [15] with default parameters. Possible human, bacterial and virus contamination was removed using BBMap V. 38.75 [16] using an alignment approach. Also, probable eukaryotic ribosomal RNA contamination was removed from raw reads using sortmeRNA V. 4.2.0 software [16]. Ribosomal 5S, 18S, and 28S RNA subunit sequences that served as a reference were downloaded from rfam V. 14.1 [17] and SILVA V. 138 [18] databases. Processed reads were checked in MultiQC V. 1.8. and used as input data for RNA-Seq *de novo* assembly and downstream analysis.

### 2.5. RNA-Seq de novo Assembly

For RNA-seq *de novo* assembly, rnaSPAdes V. 3.13.0 [19] and Trinity V. 2.8.5 [20] *de novo* and reference-guided methods were tested. rnaSPAdes assembly pipeline uses a range of k-mer values to generate optimal k-mer values for assembly automatically. For Trinity, assemblers were tested with different k-mer values. Also k-mer values from the previous *de novo* guar leaf transcriptome assembly were tested (k-mer = 25, k-mer = 32) [21,22]. Trinity *de novo* and reference-guided assemblies and rnaSPAdes V. 3.13.0 assembly were compared using Transrate V. 1.0.3 [23] based on such assembly quality characteristics as total number of transcripts, total and average length of assembled transcripts and N50. 

A preliminary guar genome ~420 Mb assembly combining short and long reads with ~300× coverage created with MaSurca V. 3.3.8 [24] was used as a reference for Trinity genome-guided method. Transcriptomic reads were aligned to the reference genome using STAR V. 2.7.2 aligner [25] and assembled with Trinity. Assembly quality was checked by aligning cleaned reads against the transcriptome assemblies using bowtie2 V. 2.4.1 [26]. To evaluate assembly quality Benchmarking Universal Single-Copy Orthologs database (BUSCO) v4.0.0. [27] was used with default E-value cut-off parameter (1 × 10^−3^) against two ortholog databases: embryophyta_odb10.2019-11-20 (1614 markers) lineage and fabales_odb10.2019-11-20 (5366 markers) from OrthoDB v10. This analysis provides us information about assembly integrity and completeness based on comparison of the assembly with orthologous gene markers from selected databases (groups). 

Clustering of transcripts was done using CD-HIT-EST from the CD-HIT V. 4.8.1 [28] with sequence identity cut-off parameter 97% and length ≤ 200.

### 2.6. Differential Expression Analysis

For the downstream analysis all of these longest isoforms were searched using blastx algorithm of blast+ V. 2.9.0 [29] against transcripts database made from *Arabidopsis thaliana* transcripts downloaded from Phytozome V. 12.1 [30] using parameters of percent identity cut-off > 70%, e-value cut-off 1 × 10^−3^ and maximum blast hits = 1. Isoforms abundance was estimated with RSEM V. 1.3.2 [31] and filtered by Trinity V. 2.8.5 [20] plugin. 

Differential gene expression was analyzed between the groups of early and delayed flowering lines using the longest isoforms of transcripts. Differential expression was assessed with the DESeq2 V. 1.26.0 R package [32] using a statistical model accounting for differences among conditions (Early VS Delayed). Using this model, the possible differences among lines within each group of early and delayed lines is kept in the residual. This choice allows us to increase specificity of the analysis, but with the possible drawback to decrease power of detecting DEGs. Normalization was done by median of ratios, as included in the DESeq2 package. The most significant transcripts were filtered and saved based on adjusted *p*-adjusted values < 0.05. 

### 2.7. Enrichment of Differential Expressed Genes (DEGs)

To obtain a first insight into the biological meaning and functions of the differentially expressed transcripts, gene set enrichment analysis (GSEA) was performed using the Clusterprofiler V. 3.12.0 R package [33]. For this purpose all transcripts were ranked according to their log_2_fc values and compared to gene lists from org.At.tair.db database v 3.12 from the Bioconductor package for GO-ontology analysis. 

To confirm whether the differentially expressed genes belong to functional categories related to flowering processes we performed a Gene Set Enrichment Analysis (GSEA). The 11,684 DE transcripts were first assigned into different functional categories defined for *A. thaliana* (org.At.tair.db V 3.12, https://bioconductor.org/packages/release/data/annotation/html/org.At.tair.db.html Accessed on 15 June 2021 ) via GO-ontology.

### 2.8. Relationship between Metabolites and Transcripts: Shiny GAM Network Construction

Shiny GAM online software (https://artyomovlab.wustl.edu/shiny/gam/ Accessed on 15 June 2021) [34] was used for transcripts and metabolites integration based on graph scoring. For this approach, 63 metabolites mapped to KEGG database and 10,663 of the DE transcripts with a BLAST hits against *Arabidopsis* transcripts from Phytozome V. 12.1 database that satisfied the criteria of base mean expression >10 (10 reads per transcript) were used.

Network construction was performed in the following way: differential expression for genes was converted to differential expression for reactions. This was performed by converting all genes that code enzymes to a biochemical reaction that they take part in. The gene with the minimal *p*-value was selected and its *p*-value was assigned as the reaction *p*-value. All reactions without *p*-values were discarded as having no expressed enzymes. Reactions in our network were interpreted as nodes. For downstream gene network analysis groups of reactions that have at least one common metabolite and the same most significant gene were collapsed into single nodes. The network construction was performed under default parameters. 

## 3. Results

### 3.1. Gas Chromatography–Mass Spectrometry Metabolomic Analysis

Metabolite profiling by GC-MS was performed as previously described, repeating the same 2018 experiment with the same guar population, each genotype represented with at least three technical replicates [7]. As a result, 98 peaks were detected for all samples. Analysis using GMD and NIST libraries identified 69 metabolites. These metabolites were merged with the KEGG database (https://www.genome.jp/kegg/pathway.html Accessed on 15 June 2021) *Arabidopsis* IDs and, finally, a matrix of 63 metabolites was obtained (Appendix A).

Comparing results of two years of experiments (2018 and 2019), we found that the metabolome detected for the same guar genotypes differs significantly depending on the growing conditions. In the experiment of each year, the plants were grown under the same natural conditions of lighting, humidity and temperature. However, there were differences between the growing conditions in the 2018 and 2019 experiments. Consequently, the same plants at the same developmental stage reached different heights and accumulated different biomass in 2018 and 2019 (Appendix A).

As a result, the number of metabolites, whose concentrations differed significantly in early flowering (EF1) and delayed flowering (DF1) plants, varied from 65 (2018) to 36 (2019), having 14 metabolites in common. Eleven of them were significant by FDR criteria for the both years. Among them there were five key metabolites (myo-inositol, chiro inositol, tetronic acid, cinnamic acid, liquiritigenin), which we proposed earlier as biomarkers of early flowering in guar [7].

### 3.2. RNA Sequencing and Quality Control 

Fifteen cDNA poly-A enriched libraries were sequenced with NovaSeq6000 to generate 4.1 Gbp of raw single-end reads. Each cDNA library corresponded to approximately 26–30 million single-end reads. After all filtering steps ~80% of initial reads per sample were retained for downstream analysis (Appendix A).

### 3.3. RNA-Seq de novo Assembly 

RNA-seq *de novo* assembly was performed using rnaSPAdes V. 3.13.0 [19] and Trinity V. 2.8.5 [20] *de novo* and reference-guided methods. The comparative statistics of the assemblies were obtained by Transrate V. 1.0.3. (Table 2). From 102,539 contigs with mean length of 936 bp to 132,825 transcripts with mean length of 539 bp were generated using genome-guided Trinity assembly and rnaSPAdes assembler with default k-mer values, respectively. N50 value ranged from 1394 for rnaSPAdes to 1661 Trinity genome-guided assembly. 

All assemblies were subjected to Benchmarking Universal Single-Copy Orthologs (BUSCO) analysis to assess completeness of assembled transcriptomes. Two datasets of single-copy orthologs (*Embryophyta* and *Fabaceae*) were used. Trinity genome-guided assembly which had the highest number of BUSCO complete groups compared to other assemblies (90.8% for *Embryophyta,* 82.4% for *Fabaceae*) was selected for further analysis (Appendix A).

Trinity genome guided assembly contains 102,539 contigs corresponding to 79,863 unigenes (Table 2). Unigenes represent the number of unique assembly contigs or scaffolds located in the same isoform cluster. To improve the accuracy of the unigenes clusterization, all 102,539 transcripts from Trinity genome-guided assembly were clustered using CD-HIT. In total, 96,447 clusters of all revealed isoforms were retained.

So far, several transcriptome assemblies for guar were reported in literature: among them one specific to roots [35] and two specific to leaves [21,22]. The latter studies focused on leaf tissue which was picked up from 21-day-old seedlings. In our experiment leaf material was collected on the 48th day after sowing. At this particular developmental stage, less photoperiod-sensitive genotypes have accumulated all the necessary metabolites to start the flowering process, while genotypes sensitive to the photoperiod have not. Overall, transcriptome obtained in this study is comparable with assemblies previously reported for guar (Table 3). 

### 3.4. Differential Expression Analysis

Out of 96,447 clusters of isoforms obtained after CD-HIT clusterization, only 78,015 longest isoforms were kept for the further analysis of differential expression. These were equally distributed across all 15 guar libraries under analysis. The mean number of transcripts per library varied from 225 up to 348 with the average raw counts depth for all samples of 264 (Appendix A). 

PCA biplot based on all the 78,015 longest isoforms shows the repartition of the fifteen guar plants belonging to five distinct lines. Figure 1 shows that Early and Delayed flowering genotypes can be readily separated.

Differential gene expression was estimated using RSEM and DESeq2 R packages between groups of Early Flowering (EFl) and Delayed Flowering (DFl) guar lines. The first group consisted of the nine plants belonging to lines 34, 69, 97 and the second group combined the six plants belonging to lines 75 and 28. 

11,684 out of the 78,015 transcripts were successfully annotated with *A. thaliana* BLAST analysis against the Phytozome database and kept for further analysis. Significant differentially expressed (DE) transcripts were identified based on adjusted *p*-value < 0.05. *P*-values were uniformly distributed with a small deviated fraction of DEGs with *p*-values < 0.05 (Appendix A). Negative log_2_fc values indicated decreased gene expression in EFl genotypes when compared with the global mean. Respectively, positive log_2_fc values mean a gene overexpression in EFl plants. 

As a result, a list of 1067 DE transcripts was obtained, combining 533 overexpressed and 534 down-regulated transcripts in Early flowering genotypes in comparison to Delayed flowering ones (Appendix A). 

### 3.5. Gene Ontology (GO) Enrichment Analysis

Guar transcripts were assigned into functional categories of *A. thaliana*. The 11,684 blasted DE transcripts classified to 64.41% of guar transcripts were attributed to biological processes (BP) known for *A. thaliana,* 55.6% to molecular functions (MF) and 68.03% to cellular components (CC). 

Figure 2a suggests that the most of the biological pathways suppressed in Early flowering plants are related to response to abiotic stimulus, like temperature and light reactions. Among them were genes regulating photoperiod pathways, e.g., ELF6 encoding zinc finger domain-containing protein (*AT5G04240*), photoreceptors (e.g., *CKL4:AT4G28860, JAC1:AT1G75100*, *SPA1:AT2G46340*, *ckl3:AT4G28880*, *PHOT2:AT5G58140*), response to heat (*GFA2:AT5G48030*), (Appendix A). 

On the other hand, floral organ development pathway is expectedly activated only in Early flowering plants by overexpression of genes responsible for gynaeceum development and carpel formation (*HEC1:AT5G67060*) and regulation of shoot apical meristem and leaf development (*AFO:AT2G45190*). The *AGL6* gene, which encodes a MADS-box transcription factor and positively regulates *FLOWERING LOCUS T (FT)* in *Arabidopsis,* was also overexpressed in EFl guar genotypes. Accordingly, several biological processes related to transcription and RNA biosynthetic processes were activated in EFl plants.

Apart from the genes whose overexpression or down expression in EFl genotypes can reasonably be assumed, there were also DE transcripts belonging to common and housekeeping pathways. The possible reason is that some genes can be classified in biological processes as related to floral organ development and at the same time were assigned by their molecular function to DNA binding or DNA binding transcriptional factor activity, e.g., *HEC1:AT5G67060*, *AFO:AT2G45190*, *AGL6:AT2G45650*, *KNAT2:AT1G70510*, *INO:AT1G23420*, *SPL9:AT2G42200*, *SEP1:AT5G15800*.

### 3.6. Integrative Approach for Metabolites and Transcripts Analysis Using Shiny GAM Network Application

Shiny GAM software was used to construct a gene network allowing the integration of selected transcripts and metabolites. 10,663 guar transcripts, represented by the longest isoforms which were successfully aligned with BLAST against *A. thaliana* transcriptome, with the number of reads for each transcript >10, were uploaded into the Shiny GAM network. On the other hand, the metabolite matrix containing concentrations of 63 metabolites was used as metabolite input. Not all the metabolites could be implemented in the network, as some metabolites were involved in different biosynthesis pathways and connected with the different genes, appearing in the plot several times. As a result, a gene network with 98 nodes (metabolites) and 125 edges (genes) was constructed (Figure 3).

In the Shiny GAM network one can see 16 significant metabolites, 15 of them showed significantly higher concentration in EFl guar plants as compared to DFl plants (Figure 3). 9 out of 16 metabolites integrated into the gene network were connected with significant DE genes encoding enzymes involved in the nearest biochemical reaction (Table 4). Of greatest interest were the “key” metabolites detected in both the 2018 and 2019 studies examining metabolites associated with flowering time in guar and suggested as biomarkers of early flowering. 

Lactose was the only metabolite with negative log_2_fc in early flowering genotypes (Table 4). This metabolite was connected with *BGAL17:AT1G72990* gene that encodes beta-galactosidase 17 involved in lactose hydrolysis. *BGAL* is one of the glycosidases found to have increased activity during seed germination of legume species. For example, β-Galactosidase activity was reported during the course of mung bean germination, where it possibly participates in the mobilization of cell wall polysaccharides: the enzyme activity increased from day 1 after imbibition, approached a high level around day 5 and remained about the same until day 9 [36]. Since expression of the β-galactosidase genes are subjected to developmental regulation, we hypothesized that at the stage of the third true leaf in rapidly developing EF1 plants, the mobilization of cell wall polysaccharides in cotyledons has already been completed, which led to a decrease in the expression of the *BGAL17:AT1G72990* gene as well as the concentration of lactose. On the other hand, there is strong evidence that in *Arabidopsis BGAL* genes are regulated by abiotic and biotic stresses [37].

#### 3.6.1. 4-Coumarate (Cinnamic acid)

4-Coumarate is the main substrate for the CoA ligases involved in the biosynthesis of flavonoids and many other low molecular weight phenylpropanoids, as well as in guaiacol lignin formation [38]. Thus, suppression of the *4CL3:AT1G65060* gene encoding 4-coumarate CoA ligase is expected to be associated with the higher concentration of 4-Coumarate metabolite, since it is responsible for conversion of 4-Coumarate into p-coumaroyl CoA [39]. Besides flavonoid biosynthesis, 4-coumarate CoA ligase (*4CL*) together with the phenylalanine ammonia-lyase (*PAL*) enzyme is involved in the phenylpropanoid pathway in plants. Genes encoding these enzymes are coordinately activated in response to developmental cues [40]. In the Shiny GAM network two genes from the phenylpropanoid pathway *4CL3:AT1G65060* (4-coumarate CoA ligase isoform 3) and *PAL2* were down-regulated in EFl plants resulting in expected higher accumulation of the 4-Coumarate metabolite.

#### 3.6.2. D-Glycerate

GLYK (D-glycerate 3-kinase) catalyzes the last reaction of the photorespiratory C2 cycle and is one of the core enzymes of plant carbon assimilation. This enzyme converts glycerate into 3-phosphoglycerate (3PGA) during the final step of photorespiration in the chloroplast [41]. Thus, downregulation of the *GLYK:AT1G80380* gene could explain the high concentration of D-Glycerate metabolite detected in EFl guar plants (Figure 3). *AT1G80380* which encodes GLYK in *Arabidopsis,* displays phytochrome-regulated alternative splicing probably due to changes in alternative promoter selection: a long mRNA encodes plastid-localized GLYK (ptGLYK) in the light, and a short mRNA encodes cytoplasmic GLYK (cytGLYK) with truncation of the N-terminal transit peptide in darkness. Because we choose to include the largest isoforms in the integrative analysis, we assume that the long plastid-localized *GLYK* transcripts, expected to be overexpress in EFl plants in the light, are present in the Shiny GAM network. 

#### 3.6.3. Citrate

*ACL* (ATP-citrate lyase) is located in the cytosol and responsible for generating the cytosolic pool of acetyl-CoA by catalyzing the ATP- and CoA-dependent cleavage of citrate [42]. The *ACL* antisense RNA *Arabidopsis* plants exhibit delayed flowering compared to wild-type plants. *Arabidopsis* flowering induction is highly sensitive to alterations in *ACL*-derived acetyl-CoA metabolism. Reductions in ACL activity to 35% of wild-type levels generate a pronounced altered phenotype; such plants do not flower and thus cannot reproduce [43]. In our experiments citrate metabolite was overrepresented in EFl guar plants and the connected *ACLA-1:AT1G10670* gene was also upregulated. Since ATP-citrate lyase catalyzes acetyl-CoA biosynthesis from citrate, the synthesis of the latter is maintained at a high level in early flowering plants.

#### 3.6.4. S-Malate

Shiny GAM connected down regulation of *NAD-ME1:AT2G13560* gene with the high concentration of S-Malate in EFl plants. The NAD-dependent malic enzyme 1 (*NAD-ME1:AT2G13560*) converts (S)-malate to pyruvate and CO_2_. Recently, *NAD–ME1* gene was reported as the most likely candidate gene underlying the *Met.II.15* QTL in *Arabidopsis* leading to altered regulation of several glucosinolate (GSL) biosynthesis pathway genes in a time-dependent manner [44]. It was showed that *NAD–ME1:AT2G13560* underlies a complex regulatory network dependent upon the day–time. Moreover, *NAD–ME1:AT2G13560* co–expressed with CRY2 (Cryptochrome 2) and PHYA (Phytochrome A), which are the key components of the circadian oscillator complex [45]. 

#### 3.6.5. Myo-Inositol

One of the key metabolites, namely myo-inositol, was described earlier as a biomarker of early flowering in guar [7]. Downregulation of a gene (*ATGOLS1:AT2G47180*) encoding galactinol synthase is associated with the higher concentration of myo-inositol in EFl plants (Figure 3). Galactinol synthase is a key enzyme in the raffinose biosynthesis pathway, catalyzing synthesis of galactinol from UDP-galactose and myo-inositol (GolS or GAS, EC 2.4.1.123). It looks logical that a down expression of the enzyme leads to an increase in myo-inositol, so the pattern of regulation of the galactinol synthase gene and the concentration of myo-inositol in guar plants as established by Shiny GAM looks consistent. It was suggested that myo-inositol may participate in cold-induced transcription of the gene in *Medicago falcata* (*MfGolS1*) providing multiple tolerances to abiotic stresses [46]. On the other hand, in *Arabidopsis* the seed germination was faster in knockout mutants targeting *ATGOLS1:AT2G47180* compared with wild-type plants, suggesting that *Arabidopsis* galactinol synthases 1 (*ATGOLS1::AT2G47180*) negatively regulates seed germination [47]. As we reported earlier, the flowering time in guar depends on genetic factors that determine the speed of seed germination and the formation of the first true leaf [6]. Thus, we assume that a difference in expression of an *ATGOLS1:AT2G47180* gene may contribute to the difference in flowering time between EFl and DFl genotypes. 

### 3.7. Myo-Inositol as a Biomarker of Flowering Time

To reveal other possible links between myo-inositol key metabolite and genes underlining the Shiny GAM network, we evaluated the KEGG myo-inositol phosphorylation pathway (*ath04070*) map available for *A. thaliana*, also known as phosphatidylinositol signaling pathway.

To get an insight into the interaction of myo-inositol with up/down regulation of differentially expressed transcripts we mapped differential expression data to the biochemical pathway and colored participating genes by log_2_fc value (Figure 4). Clusterprofiler software identifies genes as differentially expressed based on log_2_fc only, not taking into account their *p*-value. Therefore, not all DE genes colored on the phosphatidylinositol signaling pathway map (Figure 4) have a *p*-adjusted value < 0.05 (Table 4).

Six enzyme entries were overexpressed in EFl guar plants (namely, 3.1.3.25, PLC, PTEN, 3.1.3, 2.7.8, 3.1.3.57, colored by red on Figure 4), while six enzyme entries were down-regulated (namely, PIKFYVE, PI4K, PIP5K, 2.7.1.107, 2.7.4.24, 2.7.1.59 marked by green). When enzymes with the highest or lowest log_2_fc values from these two groups were analyzed, a remarkable pattern emerges (Table 5). Almost all the genes overexpressed in EFl guar plants (positive log_2_fc) encoded phosphatases, catalyzing dephosphorylation of myo-inositol phosphates (InsPs). The latter are presented in the phosphatidylinositol signaling pathway by metabolites of variable phosphorylation on a carbohydrate core - myo-inositol (marked by circle on Figure 4). Conversely, all the down-regulated genes (negative log_2_fc, Table 5) encode phosphate kinases, responsible for myo-inositol phosphorylation. On the metabolite chromatograms obtained by GC-MS-metabolomic analysis a clear distinctive peak was assigned to myo-inositol in its free, dephosphorylated form. Thus, the recorded high concentration of this metabolite in EF1 plants could now be explained by the orchestral suppression of myo-inositol phosphorylases and alternative overexpression of phosphatases, which catalyze InsP hydrolysis by cleavage of phosphate.

Among the six overexpressed genes there was *phospholipase C* (*PLC:AT5G58670*) - the enzyme playing an important role in signal transduction pathways. *PLC* hydrolyzes phosphatidylinositol-4,5-bisphosphate (PIP2), a phospholipid that is located in the plasma membrane, releasing inositol 1,4,5-trisphosphate (InsP3), a key signaling molecule. We could not detect the ‘heavy’ InsP3 metabolite and other inositol phosphates in our analysis using GC-MS approach, the high-performance liquid chromatography (HPLC) should be used instead. Nevertheless, the activation of *PLC:AT5G58670* genes in EF1 plants indicates that the signaling pathway was initiated, and a high concentration of the free form of myo-inositol can serve as a marker of the onset of this process.

## 4. Discussion

In the present study we employed ‘omics’ technologies and systems biology, aiming to provide solutions to a key issue that arises in breeding practice. How to manage the genetic determinism of flowering time is often a first critical step when introducing a new agricultural crop into unusual environmental conditions. Flowering time is a key agrobiological trait for crops and dissection of genes controlling the flowering phenotype is one of the major issues of plant geneticists. In this study we establish a link between the metabolome landscape, accompanying transition of guar plants to the generative stage, with the transcriptome profiling. 

The transcriptome analysis of three early flowering and two delayed flowering guar lines (using respectively nine and six cDNA libraries from individual plants) was done using Illumina NovaSeq6000 technology. Approximately 25 million single end reads from each sample were used to generate *de novo* transcriptome assembly for guar that contained 102,539 contigs. We used a genome-guided transcriptome assembly approach, employing results of our previous study [48]. That allowed us to identify the largest number of unigenes published for guar so far. However, since a reference annotated genome is not yet available for guar, the consequent differential expressed study faced some challenges.

Only 15% of 78,015 identified guar transcripts were successfully annotated via *A. thaliana* BLAST and could be used for subsequent DE analysis. As a result, 533 genes were found overexpressed in Early flowering lines as compared to Delayed flowering lines, and 534 were down-regulated correspondingly. Gene set enrichment analysis showed that among the overexpressed genes were those involved in the pathways of floral organ development. This indicates that the developing stage of the third leaf of guar is the correct stage for identifying the genes responsible for the onset of flowering. 

Many DE genes, showing high log_2_-fold change value did not pass the adjusted *p*-value < 0.05 threshold probably due to significant variation between genotypes within the Early and Delayed phenotype group. For example, the Flowering Time gene (*FT:AT1G65480*), which is supposed to play a role in distinguishing between EF1 and DF1 plants, showed a high value of log_2_fc = 2.4, but not a significant value of *p* = 0.1918. 

A higher sample size could remove this discrepancy. However, it should be taken into account that in a sample of unrelated genotypes the early flowering phenotype can be achieved in different ways; therefore, it is not at all necessary that a particular candidate gene be expressed in the same way in all analyzed early flowering plants. That is one of the reasons why the transcriptome analysis has been supplemented with metabolome profiling.

We previously suggested that a particular metabolomic landscape is established when a guar plant is ready to switch to flowering. Seven key molecules were proposed as potential biomarkers of early flowering in guar genotypes on a stage of the third true leaf [7]. Here, we confirm that there are key metabolites whose high concentrations in tissues of the third true leaf indicate the upcoming onset of flowering, even if the metabolome profile detected by GC-MS approach can vary depending on growing conditions.

Next, we used an integrative approach to understand the gene expression network underlying the metabolic landscape that accompanies the onset of flowering in guar. Shiny GAM network application links a gene expression to the nearest biochemical reaction product. This integrative analysis was only available for those guar transcripts that were successfully blasted against *A. thaliana* transcriptome, which means 15% of the 78,015 transcripts detected. On the other hand, some of the guar metabolites were not included in the integrative analysis, since the corresponding transcripts were not available in the *Arabidopsis* gene network. Finally, many of the guar metabolites successfully integrated into the Shiny GAM network showed no significant difference between EF1 and DF1 plants. As a result, we were able to analyze genes associated with only 16 significant metabolites, six of which were overrepresented in EF1 plants compared to DF1 plants in two-year replicates of the metabolome profiling experiment. 

In some cases, as revealed by Shiny GAM network, a higher concentration of the metabolite was associated with the suppression of the enzyme for which the metabolite served as a substrate. On the other hand, for the key metabolite D-Glycerate an interesting finding was reported about the connected *D-glycerate 3-kinase* gene, that is proved to regulate by the *phytochrome phyB*, depending on the light conditions. Assuming that flowering time can be affected by the photoperiod sensitivity of guar genotypes, the polymorphism of the *phyB* gene between EFl and DFl plants can be hypothesized and may be assessed in follow-up work. The most interesting finding was the connection of *S-Malate* metabolite and the NAD-dependent malic enzyme 1 (*NAD-ME1:AT2G13560*). In *Arabidopsis* the gene was recently reported as the most likely candidate gene for metQTL, regulating several secondary metabolites biosynthesis pathway genes in a day-time-dependent manner coexpressing with the circadian oscillator complex [44]. Thus, a polymorphism of the *NAD-ME1:AT2G13560* gene, affecting its expression in the EFl and DFl plants can be hypothesized.

The main finding of our metabolites-transcripts integrative study is the revealed pattern of the phosphatidylinositol signaling pathway that is presumably initiated in EFl plants ready for flowering due to the activation of the phospholipase C (*PLC:AT5G58670*) gene and leads to an exponential increase in the amount of myo-inositol in its free form. This free form of myo-inositol is detected in our GC-MS chromatograms. The signaling pathway is performed by suppression of at least five *myo-inositol phosphate kinases* (phosphorylation) and alternative overexpression of three phosphatases (dephosphorylation) (Table 5). The interconversion of the phosphorylation states of Phosphoinositides (PIs) by specific kinases and phosphatases, followed by recruitment of PI-specific effectors, is precisely the key mechanism by which the spatiotemporal regulation of PI-mediated biological processes (e.g., receptor-mediated signaling, vesicular traffic, cytoskeleton rearrangement, and regulation of channels and transporters) is achieved [44]. 

Myo-inositol was proposed earlier as a key biomarker of the early flowering in guar [7]. The higher concentration of this metabolite can be detected in tissues of the third true leaf – the developmental stage that precedes first floral bud appearance [7]. In this study, we took one step forward to understand the biochemical processes underlying the higher myo-inositol concentrations in early flowering guar plants. The next step must be taken to understand which external stimuli trigger the phosphatidylinositol signaling pathway in guar under the long day conditions and what kind of genetic polymorphism influences differential expression of *PLC* gene in early and delayed flowering plants. Our study suggests that metabolome and transcriptome information taken together, provide valuable information about biomarkers that can be used as a tool for marker-assisted breeding, metabolomics and functional genomics of this important legume crop.

## Figures and Tables

**Figure 1 genes-12-00952-f001:**
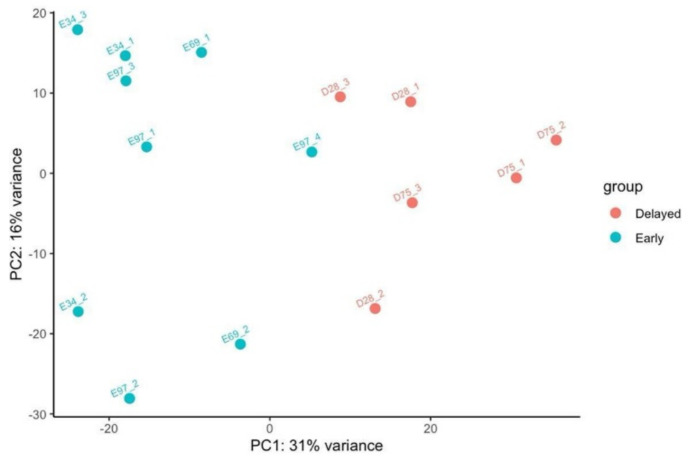
PCA biplot showing coordinates of early and delayed flowering guar lines according to the expression level of 78,015 transcripts which corresponds to the longest isoforms. For each line, three different plants were assessed for gene expression level. Plant ID (Table 1) is prefixed with E (Early) or D (Delayed).

**Figure 2 genes-12-00952-f002:**
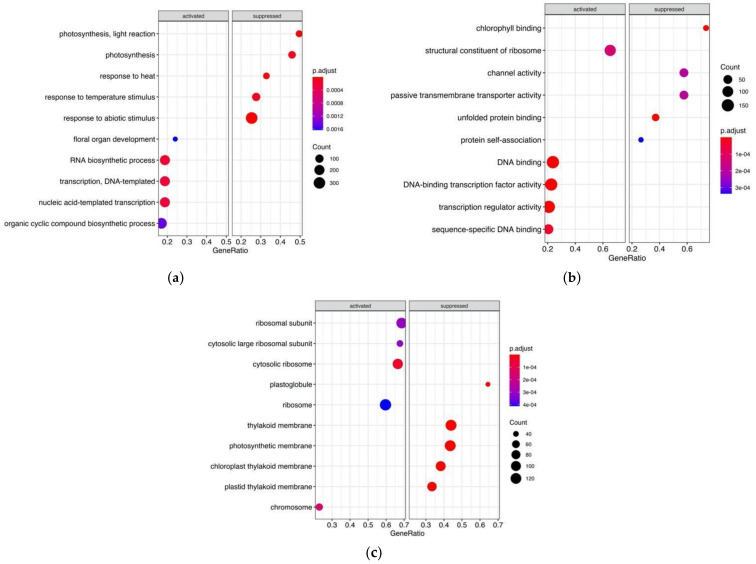
Clusterprofiler GO enrichment results showing activation or suppression of genes in Early flowering guar genotypes in comparison with Late flowering genotypes for the functional categories: (**a**) Biological Process (BP); (**b**) Molecular Function (MF); (**c**) Cellular Component (CC).

**Figure 3 genes-12-00952-f003:**
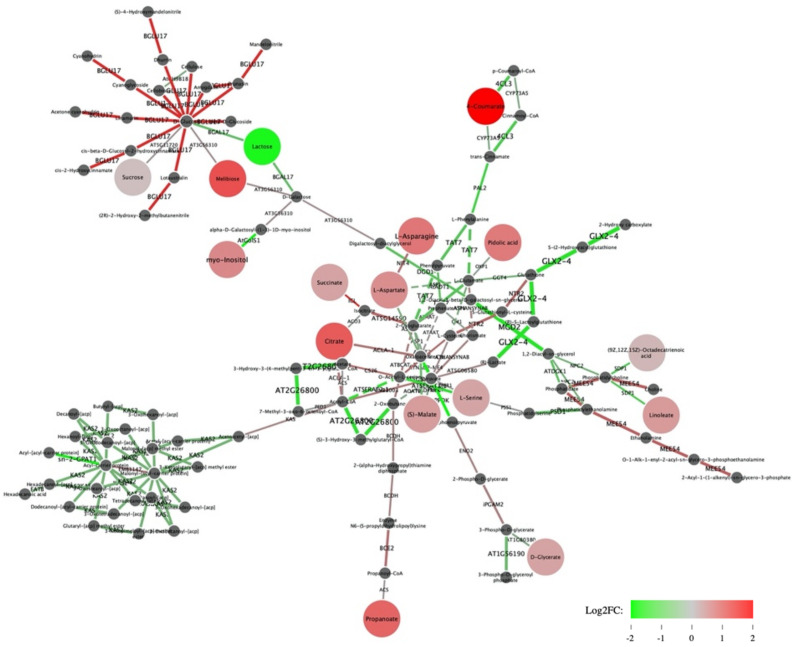
Transcriptome-metabolome network obtained with Shiny GAM. Green edges correspond to genes downregulated in Early flowering plants, red edges correspond to overexpressed genes. Green nodes correspond to metabolites whose negative log_2_fc value means lower concentrations in EFl plants compared to DFl plants, red nodes are metabolites ‘overexpressed’ in EFl. The nodes with increased size correspond to metabolites with significant *p*-value in 2019. Dark grey small nodes indicate metabolites with *p*-value > 0.05.

**Figure 4 genes-12-00952-f004:**
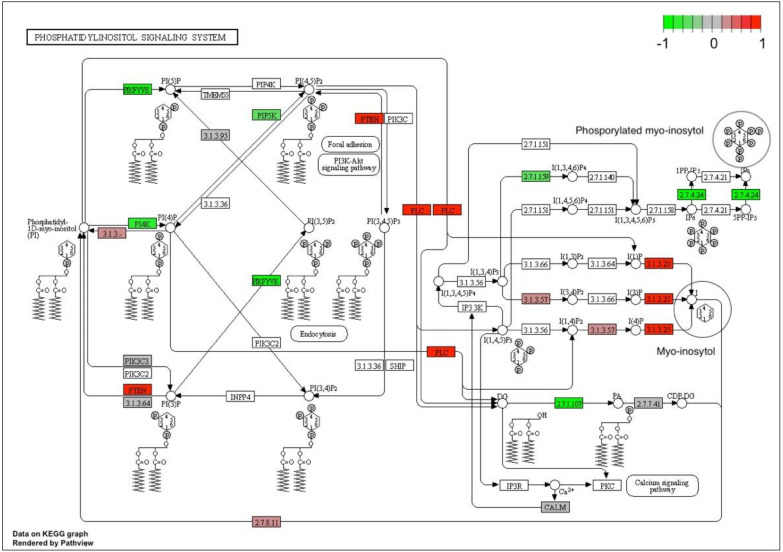
Putative transcripts involved in the phosphatidylinositol signaling pathway in guar based on homology with *Arabidopsis thaliana* on KEGG map obtained by Сlusterprofiler (pathway *ath04070)*. The red-green scale is the mean value of log_2_fc. Green labels correspond to down-regulation in early flowering genotypes, red labels to up-regulated transcripts, grey labels to transcripts with no significant difference between conditions. The target metabolite is unphosphorylated myo-inositol marked by a circle.

**Table 1 genes-12-00952-t001:** Guar lines with different photoperiod sensitivity that showed consistent early or late flowering time in experiments of 2018 and 2019 under the long day conditions.

Plant ID	Early or Delayed Flowering on Long Day Conditions	Line ID	VIR Cat. Number	Origin	Accession
34_1	Early	34	82	India	landrace
34_2	Early	34	82	India	landrace
34_3	Early	34	82	India	landrace
97_1	Early	97	52,586	USA	cv. Lewis
97_2	Early	97	52,586	USA	cv. Lewis
97_3	Early	97	52,586	USA	cv.Lewis
97_4	Early	97	52,586	USA	cv.Lewis
69_1	Early	69	52,585	USA	cv.Kinman
69_2	Early	69	52,585	USA	cv.Kinman
28_1	Delayed	28	550	India	landrace
28_2	Delayed	28	550	India	landrace
28_3	Delayed	28	550	India	landrace
75_1	Delayed	75	52,580	Pakistan	landrace
75_2	Delayed	75	52,580	Pakistan	landrace
75_3	Delayed	75	52,580	Pakistan	landrace

VIR: Vavilov Institute of Plant Genetic Resources.

**Table 2 genes-12-00952-t002:** Results reflecting assemblies’ parameters obtained by Transrate.

Metrics	Results			
	rnaSPAdes	Trinity *de novo* 32-mer	Trinity *de novo* 25-mer	Trinity Genome-Guided
Number of contigs	132,825	102,909	112,788	102,539
Shortest contigs (bp)	131	197	201	186
Longest contigs (bp)	40,377	59,908	59,908	14,406
Number of bases (bp)	78,453,910	91,396,199	103,168,619	96,000,969
Mean length (bp)	539	888	914	936
Number of contigs over 1000 bp	22,143	307	32,499	32,819
Number of contigs over 10,000 bp	95	29,453	229	3
Mean ORF %	56	54	54	54
N90 (bp)	317	324	343	357
N70 (bp)	792	862	901	980
N50 (bp)	1394	1586	1615	1661
N30 (bp)	2243	2427	2404	2359
N10 (bp)	16,753	4939	4231	3634
GC%	40	41	40	39
Mean of overall alignment rate of the transcripts against assembly (%)	96	97	99	96

**Table 3 genes-12-00952-t003:** Comparative statistics of transcriptome assemblies available for guar.

Assembly Metrics	Assembly (Genome-Guided Trinity, this Study)	Assembly Tanwar et al., 2017 [21]	Assembly Al-Qurainy et al., 2019 [22]
N50	1661	1035	2552
Total unigenes	79,863	61,508	62,146
Average transcript length (bp)	936	679	1045

**Table 4 genes-12-00952-t004:** Description of significant metabolites-transcripts connection presented in Shiny GAM network.

Metabolite Name (Shiny GAM)	Metabolite log_2_fc (2019)	Metabolite *p*-Value (2019)	Metabolite *p*-Value (2018)	Connected Genes in Shiny GAM	Gene log_2_fc (2019)	Gene *p*-Value (2019)
C00243 *(Lactose)*	−1.726	0.001	nd	*AT1G72990 (BGAL17)*	−0.711	0.001
C00137 *(Myo-inostiol)*	0.670	1.55 × 10^−6^	1.12 × 10^−7^	*AT2G47180 (ATGOLS1)*	−3.017	2.6 × 10^−10^
C00811 *(4-Coumarate)*	2.243	5.62 × 10^−4^	6.53 × 10^−7^	*AT1G65060* *(4CL3)*	−1.034	2.36 × 10^−6^
C00258 *(D-Glycerate)*	0.415	0.043	0.006	*AT1G80380 (GLYK)*	−0.303	0.039
C00158 *(Citrate)*	1.119	1.36 × 10^−4^	3.45 × 10^−5^	*AT1G10670* *(ACLA-1)*	0.497	0.001
C00149 *(S-Malate)*	0.467	9.87 × 10^−5^	0.006	*AT2G13560 (NAD-ME1); AT5G58330 (NADP-MDH)*	−0.220; −0.630	0.042; 0.004
C01595 *(Linonate)*	0.493	0.003	nd	*AT5G04040 (SDP1)*	−0.9	0.055
C00049 *(L-Aspartate)*	0.621	0.009	nd	*AT2G30970 (ASP1); AT5G22300 (NIT4)*	−0.386; 0.533	0.097; 0.006
C00065 *(L-Serine)*	0.393	0.008	0.001	*AT1G55920 (ATSERAT2;1)*	−1.448	5.492 × 10^−5^

**Table 5 genes-12-00952-t005:** Description of differentially expressed guar transcripts involved in the phosphatidylinositol signaling pathway based on homology with *Arabidopsis thaliana* on KEGG map obtained by Сlusterprofiler (pathway *ath04070*).

Enzyme Entry	Gene Name	Definition (RefSeq)	*Arabidopsis thaliana* ID	Gene log_2_fc	Gene *p*-Value	MetaCyc Database Reaction
3.1.3.25	*IMPL1*	myo-inositol 1-phosphate monophosphatase	*AT1G31190*	0.1375	0.3832	1D-myo-inositol 3-monophosphate + H_2_O → myo-inositol + phosphate
3.1.3.25	*IMPL2*	inositol-phosphate phosphatase	*AT4G39120*	0.3604	0.0600	1D-myo-inositol 3-monophosphate + H_2_O → myo-inositol + phosphate
3.1.3.25	*VTC4*	L-galactose-1-phosphate phosphatase	*AT3G02870*	0.3508	0.0522	β-L-galactose 1-phosphate + H_2_O → L-galactopyranose + phosphate
*PTEN*	*PTEN2*	phosphatidylinositol-3,4,5-trisphosphate 3-phosphatase	*AT3G19420*	0.1911	0.1132	1-phosphatidyl-1D-myo-inositol 3-phosphate + H_2_O → a 1-phosphatidyl-1D-myo-inositol + phosphate
*3.1.3.57*	*SAL1*	Inositol polyphosphate 1-phosphatase	*AT5G63980*	0.2393	0.0582	D-myo-inositol (1,4)-bisphosphate + H_2_O → 1D-myo-inositol 4-monophosphate + phosphate
*PLC*	*PLC1*	Phosphoinositide-specific phospholipase C family protein	*AT5G58670*	1.9714	0.10466	1-phosphatidyl-1D-myo-inositol 4,5-bisphosphate + H_2_O → a 1,2-diacyl-sn-glycerol + D-myo-inositol (1,4,5)-trisphosphate + H+
PIKFYVE	*FAB1A*	1-phosphatidylinositol 3-phosphate 5-kinase	*AT4G33240*	−0.6137	0.0185	1-phosphatidyl-1D-myo-inositol 3-phosphate + ATP → 1-phosphatidyl-1D-myo-inositol 3,5-bisphosphate + ADP + H+
*P14K*	*ATPI4K_ALPHA*	Phosphatidylinositol 4-kinase alpha 1	*AT1G49340*	−0.2559	0.0727	1-phosphatidyl-1D-myo-inositol + ATP 1-phosphatidyl-1D-myo-inositol 4-phosphate + ADP + H+
*2.7.11.59*	*ITPK3*	Inositol 1,3,4-trisphosphate 5/6-kinase family protein	*AT4G08170*	−0.4706	0.0392	D-myo-inositol (1,3,4)-trisphosphate + ATP→ D-myo-inositol (1,3,4,5)-tetrakisphosphate + ADP + H+
*2.7.1.107*	*DGK1*	diacylglycerol kinase 1	*AT5G07920*	−0.8194	0.00438	ATP + 1,2-diacyl-sn-glycerol → a 1,2-diacyl-sn-glycerol 3-phosphate + ADP + H+
*2.7.4.24*	*ATVIP1*	diphosphoinositol-pentakisphosphate kinase; PP-IP5 kinase	*AT3G01310*	−0.9682	0.1076	ATP + phytate → 1D-myo-inositol 1-diphosphate 2,3,4,5,6-pentakisphosphate +ADP
PIP5K	*PIP5K1*	phosphatidylinositol-4-phosphate 5-kinase 1	*AT1G21980*	−0.5195	0.00026	1-phosphatidyl-1D-myo-inositol 4-phosphate + ATP → a 1-phosphatidyl-1D-myo-inositol 4,5-bisphosphate + ADP + H+

## Data Availability

All of the mentioned data available in supplementary materials.

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
