# Peer review of "Identification of Key Metabolic Pathways and Biomarkers Underlying Flowering Time of Guar (Cyamopsis tetragonoloba (L.) Taub.) via Integrated Transcriptome-Metabolome Analysis"

_genes, 2021, doi:10.3390/genes12070952_

Round 1
Reviewer 1 Report
See annotations in the manuscript

Reviewer 2 Report
General comment
---------------
This paper describes a nice dataset that is well suited to identify key metabolic pathways and biomarkers for flowering time in Guar. The authors use a variety of integrated statistical and genomic approaches to interpret RNA-Seq and metabolomic variations between a panel of Late and Early flowering genotypes. They end-up with a list of metabolites that could be used as biomarkers to predict the sensitivity to photoperiod of Guar varieties. At each step of the analysis, the authors use appropriate methods, as well as public softwares and databases. The main drawback is a lack of analysis and synthesis. As it is, the paper looks as a practical about "what to do with omics data ?" but the different types of analyses or softwares are not compared, nor hierarchised. Although, the paper need strong rewriting : figures are not explained in the Results section, one incomplete table, missing informations. I encourage the authors to work on connecting the different parts. For example, begin with the shiny GAM network building, in which all the raw collected data have been put without prior statistical analysis and that mainly uses current knowledge to connect metabolites to reactions. Then see what WGCNA can add to the interpretation, and what additional information it brings ? Finally, add the information about the differential analysis to reduce the complexity and keep only the genes that exhibit a significant pattern. The enrichment analysis could be applied to WGCNA modules.
1. RNA-Seq de novo assembly
---------------------------
The authors compare four methods for de novo assemby : rnaSPAdes, and three methods derivend from Trinity, using either 32-mer, 25-mer or genome-guided (base on a previous assembly of Guar genome). The statistics that were used to assess the quality of the assembly seem to be N50 and the different statistics provided by BUSCO. The author claimed that they chose Trinity genome-guided based on those statistics, but BUSCO results are not provided for the three other methods. I'm not a specialist but as far as I read, N50 statictics to avaluate the quality of a de novo assembly is controversial (see e.g. https://www.molecularecologist.com/2017/05/03/n50-for-transcriptome-assemblies/)
Altogether, Tables 2-3 could be in a supplementray data. Keep Table4 and add some quality statistics from BUSCO. Some explanation/summary would be welcome for non specialists :
- The assembly contains 102,539 transcripts (Table1), corresponding to 79,863 unigenes (Table4) with mean length 936pb.
- What is the difference between a unigene and an isoform ?? How many isoforms ? Did you kept only the longest isoform for each unigene ?
- among the 79,863, only 78,015 were retained for RNA-Seq analysis. Why ??
- Explain what is a BUSCO marker, a BUSCO group ? to what need the reader compare the total number of 1,614 groups searched (Table 3)
At the end of this stage, because most of the results depend on enrichment analysis against gene ontologies, statistics should be given about the coverage of the RNA-Seq data regarding different categories. For example, a comparison with the arabidopsis thaliana genome ? A stated by the authors, only a fraction of the guar genome is expected to be expressed in the 3rd growing leaf at vegetative stage.
2. Differential expression analysis
-----------------------------------
The quality control step in DEseq generally involves normalization and filtering steps. Says a word about libraries depth. How many genes have been filtered-out prior to the differential analysis ? Fig1 is nice because it says something about the quality of the data, but it is not commented in the Results section. Something should be said also about the ditributions of the pvalues (visual checking). The volcano plot is not very informative, except to say that approx 1000 unigenes have been retained as differencially expressed. The reference chosen is not clear : is it the early or the late plants ? what does "up-regulated" means ? Is it "over-expressed in Early plants ?
The main comment about this section is that the statistical model is not formally stated. In particular, it seems that there are differences between genotypes from the same flowering-time group. I'm not sure that this level of structuration has been taken into account.
3. Enrichment of DEGs
---------------------
The authors use two different methods for enrichment analysis : gfsea and clusterprofiler. fgsea recquires to select a specied and a gene set, which are not precised in the Method section. I suppose that they choose a.t. and GO-ontology, with regards to the figures presented. Why choosing two different methods ? They seem to give very convergent results. May be one among Figure 3 and Figure 4 could be presented as Supplementary Figure ?
Also, nothing is said about the number of DEG genes that matched the a.t. genome. Results from BUSCO show only 80% hits against fabacae. How many DEGs did not have a functional category assigned ?
I'm always skeptic with enrichment analyses and what to do with it. Of course, it's nice to see that functional categories like "photosynthesis" or "response to light" or "to heat" emerge. But what about "chromosome", "DNA-binding", ... Please say a word about it. As a control, is there a category that should not be present ?
At the end of the section, the authors come-up with a list of six candidate genes, involved in photosynthesis and light response that are downregulated in the Late plants. Were they the top-6 down-regulated genes ? Why chosing those ones ?
4. Integrative approach : Metabolites and RNA-Seq WGNCA
--------------------------------------------------------
This part clearly needs rewriting.
Fig 6a,b : Exactly the same legend for (a) and (b). But I suppose that (a) is RNA-Seq and (b) is metabolome. The figures are the raw figures from the R packages. Some work needs to be done to meke them more easily readable. The correlations values are too small.
Rewrite. Don't forget (i) decribe the results; (ii) interpret. Each figure has to be commented in the Result section.
* For RNA-Seq, what I understand is that 15 modules were found, among them only two did not show differences between Early and Delayed groups. A graphical representation of the patterns, at least as supplementary figures, should be provided. The authors seem to choice only four modules among the 15 for subsequent analysis. They did not say why, but I suppose that they chose the ones with the highest correlations.
* For metabolites, they found 3 modules only, but module "black" seems to encompass most metabolites ? There is no interpretation in the paper, but it is possible that no interesting/interpretable pattern emerges from the metabolome analysis without entering RNA-Seq data, which is interesting, but not said by the authors.
* Integration : as far as I understand, the authors intersect the genes found in the four selected modules with the DEGs, and retain only DEGs that fall within the four modules. I don't understand Table5. The header for line 5 is missing. It seems to be the total number of DEGs in the DEGs column, but what for the unique elements column ? Why do all DEGs fall within the four selected modules ? Was that the criteria to select the modules ? If not this is a result that needs to be commented.
The next step consists in running WGNCA with the selected unigenes and six metabolites as groups. The results are not described nor interpreted.
5. Networks
--------------
Two networks were built.
* The first one was built with CoExpNetViz from the list of RNA-Se DEGs. The authors hilight, in the network, candidates linked to flowering time from the litterature. It would have been interesting to also highlight the candidates that were found from the enrichment analysis ? This section ends-up with a list of genes. No conclusion is drawn. Was the objective to confirm the role of already known candidates ? What about the unknowns found by the study ? Why not using WGCNA to set-up the coregulation network ?
* The second one uses the shinyGAM software and integrates both transcripts and metabolites. Nodes corresponded to metabolites, and edges to genes. This part is not connected with the previous results. Instead, the authors use all 28277 transcripts (where do they come from ? I cannot see this figure in Table 2) and all 64 metabolites to build-up the network. However, at the end, only a couple of transcripts were retained by the analysis (maybe 100 ??, the edges from Figure 9). This part show that some metabolic pathways are preferentially linked to differential gene expression among Late and Early plants : in particular, the citrate pathway and the myo-innositol pathway. They end-up with a list of connected transcripts that were also highlighted in the WGCNA analysis, and a list of biomarkers (metabolites) that could be used to scrren the varieties.
6. Focus ont he myo-innositol pathway
----------------------------------------
The last section shows the myo-innositol metabolic pathway with emphasis on the relative expression change between Early and Late genotypes.
7. Conclusion
--------------
There is no discussion. The results are never put together.
Round 2
Reviewer 2 Report
I'm globally satisfied with the answers to my comments. The authors have considerably improved the description of the results and made thoughtfull choices. However, I suggest a careful re-reading of a version after acceptation of all corrections. The present version highlights the modifications but is really difficult to read. It seems OK, but i am not sure.
Minor comments :
p3, l13-16 : the sentence does not seem correct. Is it "for all eight plants per line" ?
p10 : the volcano plot is still there but no more referenced.
